# A Comprehensive Assessment of Informal Caregivers of Patients in a Primary Healthcare Home-Care Program

**DOI:** 10.3390/ijerph182111588

**Published:** 2021-11-04

**Authors:** Virginia Rodrigo-Baños, Marta del Moral-Pairada, Luis González-de Paz

**Affiliations:** 1Primary Health Care Center Les Corts, Consorci d’Atenció Primària de Salut Barcelona Esquerra (CAPSBE), C. Mejia Lequerica S/N, 08028 Barcelona, Spain; vrodrib@gmail.com; 2Primary Health Care Center Casanova, Consorci d’Atenció Primària de Salut Barcelona Esquerra (CAPSBE), Rosselló, 161, 08036 Barcelona, Spain; mdmoral@clinic.cat; 3Faculty of Medicine and Health Sciences, University of Barcelona, C. Casanova, 143, 08036 Barcelona, Spain; 4Primary Healthcare Transversal Research Group, Institut d’Investigacions Biomèdiques August Pi i Sunyer (IDIBAPS), C. Mejia Lequerica S/N, 08028 Barcelona, Spain; 5Escola Superior d’Infermeria del Mar (ESIMar), Parc de Salut del Mar. Universitat Pompeu Fabra (UPF), Doctor Aiguader, 80, 08003 Barcelona, Spain

**Keywords:** caregivers, primary healthcare, home-care services, elderly, cross-sectional studies

## Abstract

Studies of the characteristics of informal caregivers and associated factors have focused on care-receiver disease or caregiver social and psychological traits; however, an integral description may provide better understanding of informal caregivers’ problems. A multicenter cross-sectional study in primary healthcare centers was performed in Barcelona (Spain). Participants were a random sample of informal caregivers of patients in a home-care program. Primary outcomes were health-related quality of life and caregiver burden, and related factors were sociodemographic data, clinical and risk factors, social support and social characteristics, use of healthcare services, and care receivers’ status. In total, 104 informal caregivers were included (mean age 68.25 years); 81.73% were female, 54.81% were retired, 58.65% had high comorbidity, and 48.08% of care receivers had severe dependence. Adjusted multivariate regression models showed health-related quality of life and the caregivers’ burden were affected by comorbidity, age, time of care, and dependency of care receiver, while social support and depression also showed relative importance. Aging, chronic diseases, and comorbidity should be included when explaining informal caregivers’ health status and wellbeing. The effectiveness of interventions to support informal caregivers should comprehensively evaluate caregivers when designing programs, centering interventions on informal caregivers and not care receivers’ conditions.

## 1. Introduction

Unpaid care for older, dependent persons, and informal care for family members, friends, etc., has become a central issue in countries in the latest stage of epidemiological transition. Long life expectancy and low mortality rates have resulted in an increased incidence of complex chronic diseases and disabilities that affect almost all elderly people and their families. Around 13% of people aged >50 years require informal care in OECD countries [1]. In the EU-28 group, 47.9% of elderly people reported having moderate or severe difficulties with personal care and household activities [2]. In countries with a welfare state relying on the family structure, such as Spain or Italy, informal care has become part of daily life in many families [3].

The concerns of informal caregivers (ICs) have been reported from many points of view, from sociological issues [4], national regulation, and stakeholders’ views [5], to caring activities to avoid the negative effect of losing control due to informal care, and the so-called caregiver burden [6,7]. The factors associated with the caregiver’s burden have been widely studied: sex, age, socioeconomic status, educational level, ethnicity, social support, household organization, and time spent caring have been described as triggers of the caregiver burden [7,8,9].

ICs’ health and conditions based on the care-receiver disease (e.g., dementia or terminal cancer) or grouped by situation have been studied, with some studies trying to extend the conclusions to all ICs [10,11,12,13]. Studies in the general population have reported this selection bias [8,14]. However, survey designs had other problems, such as the specificity of items and lack of consistency of the answers. Another setting to access ICs is primary healthcare (PHC) because it is the gatekeeper of the healthcare system [15,16,17,18]. Most people visit a PHC center when they require any healthcare service and they allow access to all kinds of ICs that are not restricted by disease.

PHC was envisioned in the Declaration of Alma-Ata as the “first level of contact of individuals, the family and community with the national health system, bringing healthcare as close as possible to where people live and work” [19,20]. In countries with a well-established system, PHC is accessible in centers in the community (districts, villages, and towns) where citizens have an assigned family physician and a PHC nurse. Care provision in PHC is longitudinal and cares for the most prevalent diseases, conditions, and health problems [21]. PHC health professionals coordinate with other healthcare professionals, groups, and settings (i.e., hospital or disease-oriented healthcare services). While PHC professionals have a close knowledge of patients, they are uniquely positioned to identify ICs, assess their needs, and monitor their overall health and wellbeing [22,23]. In Spain, PHC centers have home-care programs, where PHC professionals periodically visit patients who, due to health or disability, cannot reach a PHC center. The PHC home program includes palliative care, psychogeriatric home care, and a diversity of clinical conditions [22,24]. PHC professionals attend to patients’ clinical needs, and in most cases, ICs help to understand the care receiver’s situation, thus enabling patient- and family-centered decisions [22].

Studies describing the characteristics of ICs and associated factors include samples of participants not representing the general characteristics of ICs, or assessments limited to an area of their health. Most papers focus on care-receiver disease or ICs’ social and psychological traits and have not examined clinical characteristics (e.g., comorbidity) that might alter ICs’ wellbeing. Therefore, having an integrated assessment of the characteristics of ICs is crucial to plan ICs support, improve ICs care, and establish priorities for their situation independently of the disease and status of the care receiver. This study aimed to assess ICs’ demographic characteristics, health status, risk factors, PHC consultations and services, and the dependency of care receivers in the PHC home-care program. Secondarily we evaluated variables and factors associated with health-related quality of life and the caregiver’s burden.

## 2. Materials and Methods

### 2.1. Design and Context

We designed a cross-sectional study in three PHC centers in Barcelona (Spain): Les Corts, Compte Borrell, and Casanova. In 2019, before the COVID-19 pandemic, 68.38% of the total population living in the PHC area were visited once by a PHC professional. Healthcare staff consisted of 70 family physicians and pediatricians, three social workers, and 63 PHC nurses [25]. Patients unable to attend the PHC center due to bad health or mobility problems were included in the PHC home-care program.

### 2.2. Participants’ Characteristics

Potential participants were ICs of patients receiving PHC home care. If there was more than one IC in a household, we selected the one who provided the most care. Inclusion criteria were: IC and patient willingness to participate, caregiving time ≥7 h per week, permanent inclusion in the PHC home-care program, and permanent residence in the PHC area. We excluded partially institutionalized care receivers (e.g., day nursing home), IC diagnosed with severe mental illness (e.g., schizophrenia or major depression), or cognitive impairment sufficient to rule out interviews or completion of self-administered questionnaires.

### 2.3. Main Outcomes, Exploratory Factors and Variables

We defined three principal outcomes: health-related quality of life (physical and mental summaries) and caregiver burden. Four groups of factors: (i) IC sociodemographic characteristics, (ii) IC health status, risk factors, PHC use, and (iii) characteristics of care receivers, (iv) IC social assessment, and perceived support were used.

#### 2.3.1. Caregiver Sociodemographic Characteristics

We collected information from ICs on sex, age, marital status (married or partner, divorced, single, widowed), country of birth, number of residents in the household, educational level (No studies, primary school, high school and vocational studies, degree and higher), employment status (unemployed, unable to work, retired, housekeeping tasks, working), social class (based on the occupation of the person of reference) [26], source of income (employed, unemployment welfare benefits, retirement or widowhood pension, disability pension, welfare payment for dependent children or family assistance, other regular social benefit or subsidy).

#### 2.3.2. Caregiver Health Status, Risk Factors, Health-Related Quality of Life, and PHC Consultations

We included information on ICs’ diagnoses from the medical record, including conditions to compute the Charlson Comorbidity Index [27] and diagnoses of anxiety, depression, cardiovascular disease (myocardial infarction, angina pectoris, coronary disease, and other heart diseases), drug dependency, chronic respiratory disease (asthma, chronic bronchitis, emphysema, and COPD), and back pain (lumbar or cervical). Five risk factors were considered using the WHO criteria [28]: high blood pressure, diabetes (types 2 and 1), dyslipidemia (elevated cholesterol, triglycerides, or both), smoking (current smoker, ex-smoker, non-smoker), risk of alcohol dependency (units of alcohol per week: males ≥27, females ≥14) [29], and active psychoactive drug prescription (benzodiazepines, hypnotics, antidepressants, anxiolytics). PHC use was measured by the number of consultations with family physicians and PHC nurses in the last year, as well as health-related quality of life with the SF-12 questionnaire [30].

#### 2.3.3. Characteristics of Care Receivers

We collected information on sex, age, functional independence in activities of daily living (Barthel Index) [31], degree of disability (measured with the Functional Independence Measure,) [32], and hospital referral during the last year (programmed admission, emergency room, both, and no hospital referral).

#### 2.3.4. Social Assessment and Perceived Support of ICs

We collected information on support from relatives, shared IC status (caring not shared, shared with relatives, shared with a non-relative, shared with a professional caregiver), financial welfare assistance, and use of the respite program. Perceived social support was studied using the DUKE-UNC 11 questionnaire [33], social risk assessment with the Gijon test [34], and the caregiver burden with the Zarit Burden Inventory [35].

### 2.4. Description of Instruments and Questionnaires

The Charlson Comorbidity Index [27] is a weighted index that predicts the risk of death within one year of hospitalization; it includes 19 diseases rated from 1 to 6, with a total score that varies between 0 and 37 points. Weights are summed and adjusted by the patient’s age: from 50 years of age, one point is added for each decade. It produces three categories (0–1 points: no comorbidity, 2 points: low comorbidity, and ≥3 points: high comorbidity).

Barthel Index [31]: This widely used instrument monitors the functional status associated with self-care. It consists of 10 performance-based items with response point-scales ranging from two to four points). Total scores are used to classify patients according to their functional independence (<20 total disability, 20–35 severe, 40–55 moderate, ≥60 mild, 100 independent).

The Functional Independence Measure (FIM) [32] measures the degree of disability, not specific to any diagnosis. The FIM includes measures of independence for self-care (sphincter control, transfers, locomotion, communication, and social cognition). The 18-item, 7-point scale summarizes the scores in two domains: the FIM motor subscale, with scores between 13 and 91, and the cognition subscale, with 5 and 35. A final summed score ranges from 18 to 126, where 18 represents complete dependence/total care, and 126 complete independence.

SF-12 [30]: A self-administered questionnaire that screens the health-related quality of life or perceived health. The response options from Likert-type scales range from three to six points and evaluate intensity or frequency. The 12 items can be summarized in two domains of quality of life: the physical component summary (PCS) and mental component summary (MCS). Both summaries’ values range from 0 to 100 points. Values above or below 50 should be interpreted as better or worse, respectively, than the reference population. Its psychometric properties have been widely reported [36].

DUKE UNC-11 [33]: A self-administered questionnaire that assesses perceived functional social support. It consists of eleven items: seven screens, confidential support (having people to communicate with), and affective support (demonstrations of love, affection, and empathy). Scores range from 11 (less support) to 55 points. The validated Spanish version is widely used [37].

Gijon test [34]: this is a hetero-administered socio-family risk assessment scale that consists of 5 items. It allows the detection of risk situations or social problems (family, economic, housing, social relations, and support from the social network). Each item scores from 1 to 5 and classifies older people at social risk or with social problems when scores are >10.

Zarit Burden inventory [38]: This self-administered questionnaire quantifies the burden of caregivers. It consists of 22 items. For each, the caregiver should indicate how often they feel, in a Likert-like range consisting of 1 (never), 2 (rarely), 3 (sometimes), 4 (quite a few times), and 5 (almost always). The scores obtained are summed, and the final score represents the caregiver burden. Scores are significantly positively correlated (*p* < 0.001) with behavioral problems in the elderly and depression scores of caregivers (R^2^ = 0.57) [39]. The Spanish version was used [40].

### 2.5. Data Collection Method

Patients in the PHC home-care program who fulfilled the participant inclusion criteria were ordered in a list using a random sequence. ICs were contacted by telephone according to the random sequence and phoned again up to three times on different days if there was no response. Potential participants were invited to participate in the study, and if they were interested, a home visit was arranged. At home, ICs were offered information orally and in writing, and if they agreed to participate, patients and ICs signed the informed consent document. Information on the variables requiring patient participation was collected at the home visit (i.e., sociodemographic data and self-administered questionnaires). In the PHC center, medical records were accessed for information.

### 2.6. Sample Size Calculation

A Spanish population study reported that 2.45% of Spanish homes had an occupant requiring informal care [8]. At the time of the study, there were 430 patients in the PHC home-care program. Assuming a prevalence of ICs of 2.45%, a random sample of 104 participants (31 ICs per PHC center) was necessary to achieve an estimated precision of 0.26.

### 2.7. Statistical Analyses

Continuous variables were expressed as mean and standard deviation (SD) or the median and 25th and 75th percentiles if there was an asymmetric distribution, and categorical variables as percentages. Descriptive statistics were applied for all factors.

Summaries of questionnaires and inventories were computed according to published instructions and directives. SF-12 values were calculated using weighting to standardize the results according to the Catalan population [36]. We computed linear regression models for quality of life, physical and mental summaries, and caregiver burden as dependent factors. Explanatory variables were ICs sociodemographic characteristics, health status and risk factors, PHC use, characteristics of care receivers, ICs support, and social assessment. The adjusted association was analyzed using three multivariate linear regression models. Variables that were statistically significant in the univariate regression models were included. First, a complete model per outcome was constructed, and then backward stepwise regression was applied using the Akaike criterion (AIC) [41]. The collinearity of explanatory variables was examined using the generalized variance inflation factor (GVIF); a GVIG <2 was considered as an indicator of no collinearity [42]. The relative importance of each variable in the three final models was examined, and the 95% confidence intervals (CI) were calculated using a bootstrap method with 1000 bootstrap samples with replacement. This method was chosen to prevent type I error because bootstrapping does not require the assumption of normality of the sample distribution. All statistical tests used an alpha level of <0.05 and 95% CI. The analysis was carried out using R software version 3.6.1 [43].

### 2.8. Ethical Approval and Consent to Participate

The study complied with the Declaration of Helsinki and followed Good Practice in Clinical Research and local laws on confidentiality and data protection. The protocol was authorized by the Research Ethics Committee of the Hospital Clinic of Barcelona (reference HCB/2016/0272). All data were anonymized.

## 3. Results

### 3.1. Caregiver and Household Characteristics

Of the 430 patients included in the PHC home-care program, 365 (84.2%) had an IC. These potential participants were contacted randomly until the minimum sample size was achieved (N = 104). The main reason for refusal to participate was no phone availability (n = 70, 27.4%) or lack of time or trust in a home visit. Figure 1 shows the inclusion process.

The mean age of ICs was 68.25 years (SD = 12.14); 54.81% were retired, 42.31% had lower supervisory and technical occupations (42.31%), and the primary source of household income was retirement or widowers pensions (69.23%). Sixty-one caregivers (58.65%) were married or had a partner, while 85.58% were caring for one adult, of whom 88.46% lived in the ICs household, and the family relationship was daughter or son in 58.65% of cases. The mean years of caring were 6.31 (SD = 4.22), with a mean weekly care time of 5.33 days (128.02 h, SD = 49.81). Ninety-five (91.35%) of ICs’ households had ≤3 dwellers. Participant descriptive characteristics and the corresponding 95% CI are shown in Table 1.

### 3.2. Caregiver Health Status, Risk Factors, PHC Consultations, Quality of Life, Social Assessment, and Social Support

Most caregivers had high comorbidity (58.65%), anxiety and depression affected 22.69% and 22.11%, respectively, and active prescription of antidepressants (20.19%) was high: benzodiazepines (25.96%) and anxiolytics (15.38%). High blood pressure (40.38%) and dyslipidemia (38.46%) affected >30% of participants, and 13.46% were diagnosed with cardiovascular disease and 30.77% with back pain. Mean consultation to family physicians was 3.78 (SD = 4.05) and to PHC nurses 9.75 (SD = 7.09). The quality of life was below the mean of the Spanish population (physical functioning = 42.87, 95% CI: 40.80 to 44.94, and mental health functioning = 40.77, 95% CI: 38.48 to 43.06) (Table 2).

### 3.3. Characteristics of Care Receivers and Caregiver Support and Social Assessment

Table 3 shows the results of IC support status and social assessment: 75% were sharing care with someone (professional, relative, or non-relative) and, when needed, were supported by their sons or daughters (53.85%). The median Zarit burden score was 35.11. However, given the Gijon test classification and the good or acceptable social situation (75.96%), only 14.42% of caregivers had used the respite program.

Characteristics of care receivers are shown in Table 4. Care receivers were mainly female (67.31%), with a mean age of 88.88 years (SD = 6.52); almost half the care receivers (n = 50, 48.08%) had severe dependence, and the FIM showed they had moderate to maximum care (median = 72.5). Moreover, 32.70% of care receivers were referred to the hospital via the emergency room in the last year.

### 3.4. Factors Associated with Caregiver Outcomes

The results of linear regression models with all variables and factors as predictors and the outcomes of health-related quality of life (physical and mental summaries), and the Zarit burden scores are shown in the Appendix A. Backward stepwise regression analysis with the Akaike criterion (AIC) resulted in the exclusion of some variables from the models. Appendix A also shows the results of this analysis. Table 5 shows the results of the final three explanatory models with selected variables. The health-related physical summary of the quality-of-life model accounted for 26% of the total variability: significant factors were chronic respiratory disease (*p* = 0.03, 19.8%) and the dependency of the care receiver (*p* < 0.05, 24.7%). The health-related mental summary of the quality-of-life model explained 43.7% of the total variability and significant predictors were social support (Duke) (*p* < 0.001, 30.7%), sex (*p* < 0.001, 23.3%), caregiver age (*p* = 0.03, 12.8%), and depression (*p* = 0.01, 23%). The caregiver burden explained 30.7% of the variability, and contributing factors were the number household residents (*p* = 0.01, 13.2%), a diagnosis of depression (*p* = 0.02, 21.4%), and dyslipidemia (*p* = 0.05, 11.4%).

## 4. Discussion

In this study, we examined all factors that might alter the health status and wellbeing of ICs’ including care-receiver characteristics. The integrated evaluation provided new insight into ICs’ health perspective. Our results show ICs had high comorbidity and health-related risk factors, including mental health issues and chronic conditions. Care receivers had moderate to severe dependency and low cognitive and motor functions. Most ICs had social support, and their quality of life and burden were associated with their physical characteristics.

Sociodemographic characteristics of ICs and care receivers reflected the aging of the population. ICs were predominantly retired and married females, with high school education. Care receivers were mostly elderly mothers with severe dependence and needed moderate to maximum care. Caring was shared with partners and sons or daughters. In a recent population study carried out in two Spanish regions, ICs were also mainly female (56.56%), caring for their mother or father (40.16%) [15]. An Australian population study reported that most caregivers were also females (64.1%) aged >50 years (43.6%), although care receivers included children with any frailty condition [14]. A population study of ICs reported that 63.1% were female. However, the mean age was slightly younger (58.01 years), and ICs were caring for parents (26.1%) and 31.5 for their partners [8]. The results showed how sociodemographic factors reflected the dramatic aging of the population, which embraces both care receivers and ICs. In our study, daughters cared for their mothers; this might be related to cultural roles and the higher life expectancy of women. Gender issues in caring for elderly people have been considered. This is relevant to public health policies intended to improve caregiving considering family circumstances and the demographic transition, and not just disease.

ICs’ high comorbidity, risk factors, and mental conditions may explain why almost half had a predicted survival of <10 years. Worse health status in ICs has been reported [14] and various studies have examined associated factors and interventions to decrease the caregiver’s burden [44,45,46]. These reports often focus on the diseases and comorbidities of care receivers, but not those of ICs. A survey of ICs not based on care-receiver disease showed that prevalence of cardiovascular diseases and endocrine disorders was not associated with the caregiver’s burden [16]. Our results suggest that ICs’ health determines the quality of life and the burden more than patient characteristics [47], except when care receivers require intense care due to total or severe dependency. Given the poor health status, clinicians, nurses, and PHC electronic medical records might include the caregivers’ health to assess whether they can carry out their tasks due to their physical status.

The results were in line with previous reports: female ICs, social support, and depression were factors conditioning their wellbeing [8,14,48]. However, a closer inspection of explanatory the models showed that sex was not a factor in the caregiver burden and the physical summary of quality-of-life models. However, the number of household residents and chronic conditions were associated conditions. Elderly caregivers still cared for even older care receivers, who were not necessarily affected by a specific disease (e.g., Alzheimer’s or other dementias). The age of ICs may be a protective factor against the burden [16,49]; however, the explanatory models showed chronic respiratory diseases and other chronic illnesses may play an essential role in ICs wellbeing. Comorbidities and the aging of ICs are not included in ICs instruction courses; therefore, the physical conditions of ICs have been left out of caregiver educational plans and research but determines their wellbeing as ICs.

Caregiver research has concentrated on care-receivers diseases [11,12,13]. A recent meta-analysis suggested that the burden of ICs based on specific care-receiver diseases made sense in two groups: mental illness and physical impairment [49]. Furthermore, the conceptual model of the caregiver burden emerged from the case of dementia, and thereafter has not been adapted (e.g., by adding more general items) to ICs’ general conditions [50]. Generalizing ICs issues from studies based on care-receivers diseases is a methodological and ethical concern. If the primary aim is to study ICs status, research might base sample calculations on IC population figures, and not generalize to all ICs. Ethically, it drives a reduction in the ICs’ circumstances to the care receiver, which produces unwished-for effects when planning studies of ICs (e.g., avoid examining the clinical status of ICs). PHC, and more specifically the PHC home-care program, allowed the study of all kinds of caregivers without depending on any disease model. Therefore, the PHC setting is probably where research on caregivers should be carried out when addressing the general status of ICs that is not disease mediated. Other settings, such as outpatient clinics and hospitals, where research on caregivers is widely carried out, are often organized by disease, resulting in their findings being biased towards the disease. In contrast, PHC nurses’ and family physicians’ consultations are patient centered (not focused on disease) and they can easily access all kinds of ICs.

The relatively limited use of social and healthcare services (i.e., respite program and financial benefits) reiterates previous evidence: public resources, including direct financial benefits, are not used by ICs [8,15,51]. Caregivers of patients included in the PHC home program might be reluctant to ask for this aid for various reasons. Older age, retirement status, and the health of caregivers might inhibit the planning of break periods. The social class of our sample might be associated with retirement pensions outside the qualification for public financial aid. Since 2006, Spain’s law has declared the universal nature of social services to help dependent citizens receive benefits [52]. However, the economic benefits for caring are below the Spanish minimum salary and most retirement pensions. Additionally, budget reductions during the last decade have taken money from most citizens’ programs; now, only the severely dependent patient can access financial aid, and in most cases, care receivers die before attaining access [53,54]. Therefore, the structural situation of caregiver programs has driven patients and caregivers to perceive exercising their legal rights as an obstacle course [5]. To sum up, PHC professionals report that poor availability, caregivers’ reluctance and refusal, mixed with guilt and refusal to acknowledge a need for a break, inhibits them even asking for help [55]. In our opinion, this issue requires a full analysis. First, caregivers’ needs must be considered, incorporating them into the decision-making process [56], which might contribute to fewer but more effective actions (programs). Second, programs designed to relieve caregivers require an evaluation system [57], specifically respite programs, which have been proven to have no effect in the short-, medium-, and long-term [58], and third, PHC professionals could set new indicators to assess the burden of caregivers effectively, mapping all resources they can easily access and working hand-in-hand with social and community stakeholders, and not be centered on the healthcare system.

This study has some limitations. The survey method may have resulted in a bias of the health status, but all participants completed all questionnaires with assistance. Diseases and diagnoses were taken from the medical record; therefore, the credibility is higher than other studies that used self-reports. As a cross-sectional study, causal links cannot be drawn. However, we did not intend this, and the models were constructed with explanatory but not predictive purposes. While the sample size ICs may seem small, the minimum sample size was achieved. Most studies with ICs include smaller sample sizes without using probabilistic methods and sample size calculations. Therefore, in our study, selection methods and sample size granted external validity to generalize results to ICs of care receivers in home-care specifically in urban areas. In our opinion, ICs from rural, suburban, and other areas might require further study. Finally, the possibility of an ICs’ decline, if selected, might pose a certain degree of bias to achieve a representative sample. However, this kind of bias also happens in all studies because participants use their autonomy rights. In our study, declining affected 19.33% of the potential participants listed. The main reason was lack of time; therefore, in our opinion, this bias does not alter the study’s internal and external validity.

## 5. Conclusions

The wellbeing of elderly caregivers of patients with high dependence is associated with aging and chronic diseases. Their perceived quality of life and the caregiver burden are explained by comorbidity and other factors such as depression and emotional support. Aging and physical conditions are issues that should be considered in programs for caregivers, while caring activities and specific caring tasks may be secondary. Health vulnerability in informal caregivers is linked with the latest stage of epidemiological transition, which will change care, and will strain the health and social systems in countries where family support and care for their relatives are central. PHC and the public health system could monitor this to maintain the effectiveness of interventions and guarantee adequate support.

## Figures and Tables

**Figure 1 ijerph-18-11588-f001:**
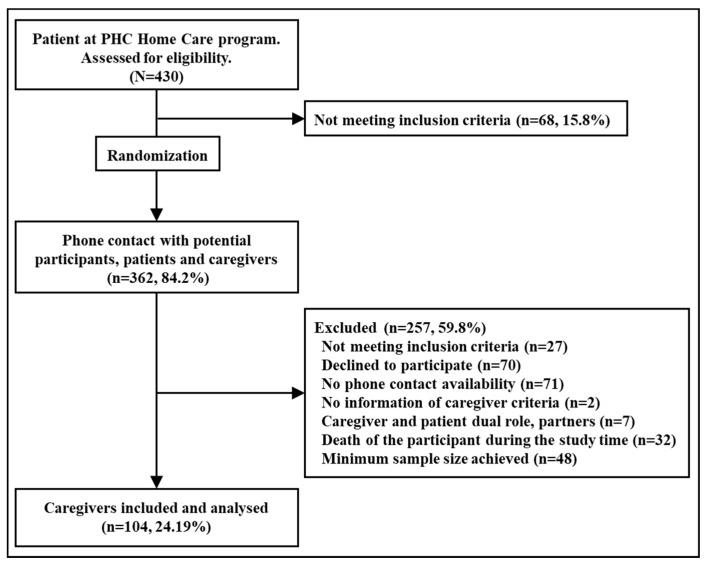
Flow chart of the inclusion process.

**Table 1 ijerph-18-11588-t001:** Informal caregiver (IC) sociodemographic characteristics.

Characteristics	N = 104 (%)	95% CI
Sex: Female	85 (81.73%)	72.95% to 88.63%
Age, mean (SD)	66.01 (12.14)	65.89% to 70.61%
Marital status		
Married or partner	61 (58.65%)	48.58% to 68.23%
Divorced	12 (11.54%)	6.11% to 19.29%
Single	22 (21.15%)	13.76% to 30.26%
Widower	9 (8.65%)	4.03% to 15.79%
Relationship with the patient		
Spouse or partner	28 (26.92%)	18.69% to 36.51%
Brother or sister	6 (5.77%)	2.15% to 12.13%
Son or daughter	61 (58.65%)	48.58% to 68.23%
Father or mother	1 (0.96%)	0.02% to 5.24%
Other relative	8 (7.69%)	3.38% to 14.60%
Employment status		
Unemployed	14 (13.46%)	7.56% to 21.55%
Unable to work	5 (4.81%)	1.58% to 10.86%
Retired	57 (54.81%)	44.74% to 64.59%
Household keeping tasks	11 (10.58%)	5.40% to 18.14%
Working	17 (16.35%)	9.82% to 24.88%
Highest educational level		
No studies	26 (26.92%)	18.69% to 36.51%
Primary studies	9 (8.65%)	4.03% to 15.79%
High school and vocational studies	38 (36.54%)	27.31% to 46.55%
Degree and higher	29 (27.88%)	19.54% to 37.53%
Social class (based on employment)		
I and II (Higher managerial, administrative, and professional occupations and lower managerial)	14 (13.46%)	7.56% to 21.55%
III (Intermediate occupations)	17 (16.35%)	9.82% to 24.88%
IV (Small employers and own-account workers)	12 (11.54%)	6.11% to 19.29%
V (Lower supervisory and technical occupations)	44 (42.31%)	32.68% to 52.39%
VI (Semi-routine occupations)	17 (16.35%)	9.82% to 24.88%
Primary source of income in the household		
Employment	23 (22.12%)	14.57% to 31.31%
Unemployment benefits	2 (1.92%)	0.23% to 6.77%
Retirement or widowhood pension	72 (69.23%)	59.42% to 77.91%
Pension for disability or incapacity	3 (2.88%)	0.60% to 8.20%
Financial benefits for dependent children or family assistance	1 (0.96%)	0.02% to 5.24%
Other regular social benefit or subsidy	3 (2.88%)	0.60% to 8.20%
The IC is the reference at their home	39 (37.50%)	28.20% to 47.53%

**Table 2 ijerph-18-11588-t002:** Caregiver health status, risk factors, PHC services use, and health-related quality of life.

IC Clinical Status	N = 104	95% CI
Comorbidity (Weighted Charlson Index)		
No comorbidity	20 (19.23%)	12.16% to 28.13%
Low comorbidity	23 (22.12%)	14.57% to 31.31%
High comorbidity	61 (58.65%	48.58% to 68.23%
Registered diagnoses and risk factors		
Anxiety	34 (32.69%)	23.81% to 42.59%
Depression	23 (22.11%)	14.57% to 31.31%
Cardiovascular disease	14 (13.46%)	7.56% to 21.55%
Drug dependency	10 (9.61%)	4.71% to 16.97%
Chronic respiratory disease	14 (13.46%)	7.56% to 21.55%
Back pain	32 (30.77%)	22.09% to 40.58%
Hypertension	42 (40.38%)	30.87% to 50.46%
Diabetes mellitus (Type 1 and 2)	22 (21.15%)	13.76% to 30.26%
Dyslipidaemia	40 (38.46%)	29.09% to 48.51%
Smoking		
Smoker	19 (18.27%)	11.37% to 27.05%
No smoker	66 (63.46%)	53.45% to 72.69%
Ex-smoker	19 (18.29%)	11.37% to 27.05%
Alcohol dependency	1 (0.96%)	0.02% to 5.24%
Active drug prescription		
Benzodiazepines	27 (25.96%)	17.86% to 35.48%
Hypnotics	7 (6.73%)	2.75% to 13.38%
Antidepressants	21 (20.19%)	12.96% to 29.19%
Anxiolytics	16 (15.38%)	9.06% to 23.78%
Health-related quality of life. SF-12 (0 to 100) mean (SD)		
Summary of physical functioning	42.87 (10.64)	40.80% to 44.94%
Summary of Mental Health Functioning	40.77 (11.77)	38.48% to 43.06%

**Table 3 ijerph-18-11588-t003:** Informal caregiver support status and social assessment.

Caregiver Support Status and Social Assessment.	N = 104 (%)	95% CI
Who provides support for care if IC cannot?		
Son or daughter	56 (53.85%)	43.80% to 63.67%
Husband/wife	35 (33.65%)	24.68% to 43.58%
Health-care professional	33 (31.73%)	22.95% to 41.58%
Brother or sister	31 (29.81%)	21.23% to 39.57%
Friends	29 (27.89%)	19.54% to 37.53%
Other relative	20 (19.23%)	12.16% to 28.13%
Neighbors	7 (6.73%)	2.75% to 13.38%
Grandson/granddaughter	5 (4.81%)	1.58% to 10.86%
Shared caregiver status		
Caring is not shared	26 (25.00%)	17.03% to 34.45%
Shared with other relative	45 (42.31%)	32.68% to 52.39%
Shared with a hired professional	34 (32.69%)	23.81% to 42.59%
Zarit burden interview. Mean (SD)	35.11 (17.16)	31.77% to 38.44%
Social Support (DUKE UNC 11). Median (IQR)	39.5 (14)	38% to 42%
Social problem or at social risk (Gijon Test)	25 (24.04%)	16.20% to 33.41%
Use of respite program	15 (14.42%)	8.30% to 22.67%
No welfare financial help to support the caring	59 (56.73%)	46.65% to 66.41%

**Table 4 ijerph-18-11588-t004:** Characteristics of care receivers.

Characteristics of Care Receivers.	N = 105	95% CI
Age, years. Mean (SD)	88.88 (6.52)	87.61% to 90.14%
Sex: Female	70 (67.31%)	57.41% to 76.19%
Functional independence in activities of Daily Living (Barthel Index)		
Total dependency	18 (17.31%)	10.59% to 25.97%
Severe dependency	50 (48.08%)	38.17% to 58.09%
Moderate dependency	36 (34.61%)	25.55% to 44.58%
Functional Independence Measure (FIM). Total. Median (IQR)	72.5 (41.5)	67 to 78
FIM, motor subscale. Median (IQR)	45 (32.5)	40 to 53
FIM, cognition. Median (IQR)	25 (14.25)	21 to 27
Hospital referral last year, reason		
Both: programmed admission and referred to the hospital emergency room	10 (9.62%)	4.71% to 16.97%
Programmed hospital admission	4 (3.86%)	1.06% to 9.56%
Referred to the hospital emergency room	34 (32.70%)	23.81% to 42.59%
No hospital referrals	56 (53.85%)	43.80% to 63.67%

**Table 5 ijerph-18-11588-t005:** Multivariate regression models of factors associated with caregiver outcomes.

Factors Associated with Caregiver Outcomes.	β	*p*-Value	% Variance Explained	Relative Importance	Bootstrap 95% Confidence Intervals (n = 1000)
**Health-related quality of life. Physical summary.**					
Time of care in hours weekly	−0.03	0.09	26.0%	16.3%	1.48% to 44.94%
Hypertension	−3.97	0.08		20.7%	3.23% to 43.13%
Diabetes mellitus	−4.64	0.08		18.5%	2.46% to 38.88%
Chronic respiratory disease	−5.96	*0.03*		19.8%	3.19% to 41.64%
Dependence of the patient				24.7%	6.38% to 53.83%
Moderate dependence	*Ref*				
Severe dependence	−6.33	*0.02*			
Total dependence	−8.02	*<0.001*			
**Health-related quality of life. Mental summary.**					
Social Support (Duke)	0.42	*<0.001*	43.7%	30.7%	10.91% to 54.20%
Sex (Male)	10.02	*<0.001*		23.3%	6.24% to 45.20%
Age	0.17	*0.03*		12.8%	1.54% to 30.25%
Depression	−6.52	*0.01*		23.0%	7.48% to 45.14%
Back pain	−3.68	0.07		10.3%	1.49% to 26.04%
**Zarit burden interview**					
Social Support (Duke)	−0.58	*<0.001*	30.7%	44.4%	15.73% to 68.39%
Number of household residents	4.05	*0.01*		13.2%	1.02% to 35.11%
Depression	8.67	*0.02*		21.4%	2.99% to 45.48%
Chronic respiratory disease	7.11	0.10		9.6%	0.37% to 34.18%
Dyslipidaemia	−5.91	*0.05*		11.4%	0.56% to 35.18%

* In italics significant *p*-values (<0.05).

## Data Availability

The data presented in this study are available on request from the corresponding author.

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
