# Peer review of "A Comprehensive Assessment of Informal Caregivers of Patients in a Primary Healthcare Home-Care Program"

_ijerph, 2021, doi:10.3390/ijerph182111588_

Round 1

Reviewer 1 Report

Congratulation! You did a big job. Please allow me some remarks.

Would you please check hyphenation, e.g. line 14.

It's a very informative abstract; however, I found it hard to understand due to the grammar.

Would you please introduce all the abbreviations add the first appearances, e.g. IC in line 31?

It's not clear how can you use the bootstrap method with n=1000 when the original population contains 430 people. (line 213)

The lover part of table 1 went to the next page.

The supporting materials are very useful

Author Response

Congratulation! You did a big job. Please allow me some remarks.

1. Would you please check hyphenation, e.g., line 14.

We have checked. We now state:

"an integral description may provide a better understanding of informal caregivers' problems."

2. It is a very informative abstract; however, I found it hard to understand due to the grammar.

We have checked the grammar.

3. Would you please introduce all the abbreviations add the first appearances, e.g., IC in line 31?

We have checked all the abbreviations, and we have added them at the first appearance.

4. It is not clear how you can use the bootstrap method with n=1000 when the original population contains 430 people. (line 213)

As a resampling technique, Bootstrap consists of replicating the sample data (n=104) with replacement a considerable number of times, n=1000 times. Resampling using Bootstrap created a proxy universe based entirely on our sample. Drawing n=1000 resamples of n=104 original sample with replacement, and we calculated the values that bound 95% of the entries or the bootstrap confidence interval.

We now state in the Methods section (line 326-330)

The relative importance of each variable in the three final models was examined, and the 95% confidence intervals (CI) were calculated using a bootstrap method with 1,000 boot-strap samples with replacement. This method was chosen to prevent type I error because bootstrapping does not require the assumption of normality of the sample distribution. All statistical tests used an alpha level of <0.05 and 95% CI

5. The lower part of table 1 went to the next page.

We have fitted this table on the corresponding page.

Reviewer 2 Report

Dear authors,

Thank you for the opportunity of reviewing your paper. The research topic is interesting: an assessment on informal caregivers. You use survey data and correctly develop a set of descriptive analysis and regression analysis. I have no major comments regarding your methodology.

However, I believe that the sample size and selection may prevent your results from being as generalizable and you claim. Additionally, it is not very clear the added-value of the paper relative to existing literature. I propose below a set of major and minor topics that, if properly addressed, might allow surpassing these two issues and improve results robustness.

In my opinion, the major issues that should be addressed are:

  1. Contribution: you claim that the concerns of informal caregivers have been reported from many points, as well as the factors associated with the caregivers’ burden. It is not clear what is the contribution of this paper to the current literature. I suggest you to clarify this in advance in the introduction (after referring to the existing literature) and in the discussion section.
  2. Sample size: further details should be given regarding the sample size. Namely why the target of achieving an estimated precision of 0.26. This estimate relies on the assumption that 2.45% of Spanish homes require informal care. Why would this be generalizable to Barcelona, and how confident are the authors about this estimate? I would expect additional discussion in this section and sensitivity analysis, as 104 participants seem an extremely low number – posing a threat to the statistical power.
  3. Participants Selection: the inclusion process does not offer enough assurance of achieving a representative sample. In fact, there is probably significant selection given that potential participants had the possibility to opt-out. The authors fail to acknowledge this key limitation. The limitation should be acknowledged and discussed. Additional details should be provided to convince the readers that the sample is in fact representative.

Additionally, some minor could be incorporated to further improve the paper:

  1. Please clarify the following sentence. Who visited who? “In 2019, before the COVID-19 pandemic, 68.38% of the total population living in the PHC area were visited once”.
  2. I suggest you to move the section on “Description of instruments and questionnaires” to the appendix, as there is no much value-added from having it in the main text.
  3. There are many typos and style problems in your sentences. I suggest you to perform a comprehensive spelling and grammar check. Eventually, it would help if your text would be reviewed by an English native speaker.

I hope you find these comments useful and that they can contribute to an improved version of your paper.

All the best

Author Response

Reviewer 2

Thank you for the opportunity of reviewing your paper. The research topic is interesting: an assessment on informal caregivers. You use survey data and correctly develop a set of descriptive analysis and regression analysis. I have no major comments regarding your methodology.

However, I believe that the sample size and selection may prevent your results from being as generalizable and you claim. Additionally, it is not very clear the added-value of the paper relative to existing literature. I propose below a set of major and minor topics that, if properly addressed, might allow surpassing these two issues and improve results robustness.

In my opinion, the major issues that should be addressed are:

1. Contribution: you claim that the concerns of informal caregivers have been reported from many points, as well as the factors associated with the caregivers' burden. It is not clear what is the contribution of this paper to the current literature. I suggest you to clarify this in advance in the introduction (after referring to the existing literature) and in the discussion section.

We now state in the introduction section (lines 71-77)

Studies describing the characteristics of ICs and associated factors include samples of participants not representing the general characteristics of ICs, or assessments limited to an area of their health. Most papers focus on care-receiver disease or ICs' social and psychological traits and have not examined clinical characteristics (e.g., comorbidity) that might alter the ICs' wellbeing. Therefore, having an integrated assessment of the characteristics of ICs is crucial to plan ICs support, improve ICs care, and establish priorities for their situation independently of the disease and status of the care receiver.

Now we state in the Discussion section (lines 285-287)

In this study, we examined all factors that might alter the health status and wellbeing of ICs' including the care-receiver characteristics. The integrated evaluation provided new insight on ICs' health perspective

2. Sample size: further details should be given regarding the sample size. Namely why the target of achieving an estimated precision of 0.26. This estimate relies on the assumption that 2.45% of Spanish homes require informal care. Why would this be generalizable to Barcelona, and how confident are the authors about this estimate? I would expect additional discussion in this section and sensitivity analysis, as 104 participants seem an extremely low number – posing a threat to the statistical power.

We appreciate this comment. We now state in the limitation section (line 605-612):

While the sample size ICs may seem small, the minimum sample size was achieved. Most studies with ICs include smaller sample sizes with no probabilistic selection methods and any sample size calculation. Therefore, selection methods and sample size granted external validity to generalize results to ICs of care-receivers at home-care specifically to urban areas. Thus, ICs from rural, suburban, and other areas might require further study.

3. Participants Selection: the inclusion process does not offer enough assurance of achieving a representative sample. In fact, there is probably significant selection given that potential participants had the possibility to opt-out. The authors fail to acknowledge this key limitation. The limitation should be acknowledged and discussed. Additional details should be provided to convince the readers that the sample is, in fact representative.

We appreciate this comment, we now state in the limitation section (line 612-630)

Finally, the possibility of ICs decline if selected might pose a certain degree of bias to achieve a representative sample. However, this kind of bias also happens in all studies because participants use their autonomy rights. In our study, declining affected 19.33% of the potential participants listed. The main reason was lack of time; therefore, in our opinion, this bias would not alter the study's internal and external validity.

Additionally, some minor could be incorporated to further improve the paper:

1. Please clarify the following sentence. Who visited who? "In 2019, before the COVID-19 pandemic, 68.38% of the total population living in the PHC area were visited once".

We now state,

In 2019, before the COVID-19 pandemic, 68.38% of the total population living in the PHC area were visited once by a PHC professional.

2. I suggest you to move the section on "Description of instruments and questionnaires" to the appendix, as there is no much value-added from having it in the main text.

We agree that description of instruments and questionnaires could debilitate readability of the text, however according to the Journal instructions this section is mandatory and should be into the main text. Therefore, we have not moved this section to the appendix.

3. There are many typos and style problems in your sentences. I suggest you to perform a comprehensive spelling and grammar check. Eventually, it would help if your text would be reviewed by an English native speaker.

 The text has been edited and corrected by an English native speaker. His contribution is under the acknowledgment section.